# Distributed Learning of Conditional Quantiles in the Reproducing Kernel Hilbert Space

**Heng Lian**
City University of Hong Kong Shenzhen Research Institute, Shenzhen, China
and
Department of Mathematics, City University of Hong Kong, Hong Kong, China
`henglian@cityu.edu.hk`

## Abstract

We study distributed learning of nonparametric conditional quantiles with Tikhonov regularization in a reproducing kernel Hilbert space (RKHS). Although distributed parametric quantile regression has been investigated in several existing works, the current nonparametric quantile setting poses different challenges and is still unexplored. The difficulty lies in the illusive explicit bias-variance decomposition in the quantile RKHS setting as in the regularized least squares regression. For the simple divide-and-conquer approach that partitions the data set into multiple parts and then takes an arithmetic average of the individual outputs, we establish the risk bounds using a novel second-order empirical process for quantile risk.

## 1 Introduction

Distributed learning has attracted significant attention recently with the increasing popularity of distributed systems and distributed data. Although the algorithmic aspects have gone far beyond the simple data parallelism paradigm that uses a simple divide-and-conquer strategy to carry out computations on each part of the data separately and then combine them in a finalizing step, the theoretical verification has lagged behind even though nowadays there are a lot of cases for which learning rates have been established, including Zhang et al. (2013, 2015); Rosenblatt and Nadler (2015); Balcan et al. (2015); Chang et al. (2017); Lee et al. (2017); Lin et al. (2017); Jordan et al. (2018); Volgushev et al. (2019); Lian and Fan (2018). It has been shown repeatedly for various models that the divide-and-conquer method can achieve the same learning rate as the standard estimator (assuming the whole data set is available on a single machine and it is computationally feasible to compute this central estimator based on the entire sample), as long as the number of partitions/machines is not too large.

In this work, we investigate the theoretical underpinning of the divide-and-conquer strategy for distributed learning of nonparametric quantile regression models in reproducing kernel Hilbert spaces (RKHS). Under this framework, the distributed learning rate for kernel ridge regression with a least squares loss has been established in several works (Lin et al., 2017; Guo et al., 2017; Chang et al., 2017). The quantile regression proposed by Koenker and Bassett Jr (1978) is an important class of models that can focus on the tail of the response distribution and is more robust the than least squares regression. However, the rate for nonparametric distributed learning other than least squares loss in the RKHS has not been established. The difficulty is that the nonparametric quantile estimator in the RKHS does not have an explicit form, which makes the decomposition of the mean squared error into bias and variance illusive, while such a decomposition is the key to establishing the rates in the divide-and-conquer method. Previously, the quantile regression estimator in the RKHS studied in Li et al. (2007); Steinwart and Christmann (2011) based on standard empirical process theory does not make it possible to decompose the error into bias and variance and thus is not adaptable to

36th Conference on Neural Information Processing Systems (NeurIPS 2022).

the distributed setting, although the *upper bound* derived there can be roughly regarded as a sum of a bias term and a variance term. On the other hand, for the fixed-dimensional parametric quantile regression, the Bahadur representation for the estimator is well-established in the literature which can be used in the distributed setting. Even in high-dimensional parametric quantile regression, Zhao et al. (2020) used a debiasing strategy so that after debiasing the bias term is sufficiently small and can be explicitly controlled. Another related work on distributed high-dimensional parametric quantile regression is Chen et al. (2020), which is more closely related to the gradient method.

To address the challenges in trying to control the bias and the variance, we use what we call the second-order empirical process involving the quantile loss function, which makes it possible to approximate more accurately the quantile loss by a weighted least squares loss, and we show that this approximation of the loss leads to approximation of the estimator (the weighted least squares estimator is close to the quantile estimator). The rest of the paper is organized as follows. In the next section, we state and prove the learning rates for both the central estimator and the distributed estimator. Section 3 contains a simple numerical illustration, which is not used to verify the learning rate (which is difficult if not impossible) but just to illustrate the divide-and-conquer method can work reasonably. The paper concludes in Section 4.

## 2 Conditional quantile estimation in the RKHS

For the response $y$, its conditional quantile at quantile level $\tau$ is denoted by $Q_{y|x}(\tau) = f_{0,\tau}(x)$. For simplicity of notation, we will suppress $\tau$ in our notations since we will focus on only an individual value of $\tau$ without modelling different quantiles jointly.

In the setting of RKHS, the unknown true function $f_0$ in the regression problem is assumed to be inside an RKHS $\mathcal{H} \subset L^2([0,1])$ with an associated kernel $K(.,.)$, with the $L^2$ space defined by a probability measure $\rho_X$ for $x$ on $[0,1]$. Note that here we assume the covariate $x \in [0,1]$ without much loss of generality. In the following we use $\langle, \rangle_{\mathcal{H}}$ and $\|.\|_{\mathcal{H}}$ to denote the RKHS inner product and the RKHS norm, respectively, while $\|.\|$ is the $L^2$-norm for functions. The function $f_0$ is estimated by

$$\widehat{f} = \arg\min_{f \in \mathcal{H}} \sum_{i=1}^{N} \rho_\tau(y_i - f(x_i)) + n\lambda\|f\|_{\mathcal{H}}^2, \tag{1}$$

where $(y_i, x_i), i = 1, \ldots, N$ is the entire sample from the true generating model with $P(y_i \leq f_0(x_i)|x_i) = \tau$, $\rho_\tau(x) = x(\tau - I\{x \leq 0\})$ is the quantile loss function, and $\lambda > 0$ is a tuning parameter. $\widehat{f}$ is referred to as the central estimator since it utilizes the entire sample.

In the distributed setting, the entire sample is distributed onto $m$ machines and each machine computes a local estimate

$$\widehat{f}_j = \arg\min_{f \in \mathcal{H}} \sum_{i=1}^{n} \rho_\tau(y_i - f(x_i)) + n\lambda\|f\|_{\mathcal{H}}^2, \; j = 1, \ldots, m, \tag{2}$$

where $(y_i, x_i), i = 1 \ldots, n$ (suppressing the dependence of the index on $j$) is the sub-sample on the $j$-th machine. We assume $n = N/m$ is an integer and $m$ is allowed to diverge with $N$. The final distributed estimator is given by $\bar{f} = \sum_{j=1}^{m} \widehat{f}_j/m$.

Let $\mathcal{L} : L^2([0,1]) \to L^2([0,1]), \mathcal{L}(f) = \int K(x,.)f(x)d\rho_X(x)$ and define $\mathcal{L}_\gamma : L^2([0,1]) \to L^2([0,1]), \mathcal{L}_\gamma(f) = \int \gamma(0|x)K(x,.)f(x)d\rho_X(x)$, where $\gamma(0|x)$ is the conditional density of $e_i := y_i - f_0(x_i)$ at 0. We will assume that $\gamma(0|x)$ is bounded away from zero and infinity for all $x$ and thus the properties of $\mathcal{L}$ and $\mathcal{L}_\gamma$ are almost identical for our purpose. We assume $\mathcal{L}$ is compact and has the spectral decomposition $\mathcal{L} = \sum_{j=1}^{\infty} s_j \phi_j \otimes \phi_j$, where $s_1 \geq s_2 \geq \cdots > 0$ are the eigenvalues and $\phi_j$ are the eigenfunctions. We assume the following smoothness condition on the true $f_0$ :

$$f_0 = \mathcal{L}^r g \text{ for some } g \in \mathcal{H}, r \in [0, 1/2]. \tag{3}$$

Assumption (3) is often referred to as the "source condition" in the literature. $\mathcal{L}^r$ is defined as the operator $\sum_j s_j^r \phi_j \otimes \phi_j$. The condition can be written more explicitly as follows. By Lemma 1.1.1 of Wahba (1990), any $g \in \mathcal{H}$ can be written as $g = \sum_j g_j \phi_j$ with $\sum_j g_j^2/s_j < \infty$, and $\mathcal{L}^r g = \sum_j s_j^r g_j \phi_j$. Thus the assumption is the same as $\sum_j f_{0j}^2/s_j^{1+2r} < \infty$, where $f_{0j} = \langle f_0, \phi_j \rangle$

is the Fourier coefficient of $f_0$ in the expansion. Since $s_j \to 0$, larger $r$ implies faster decay of $f_{0j}$, and the decay rate of the Fourier coefficient is naturally a characterization of its smoothness in functional analysis.

We establish the error bounds for the central regularized quantile estimator (2) as well as the distributed estimator. For the central estimator, although quantile regression has been studied in several works such as Li et al. (2007); Zhang et al. (2016), these works simply assume $f \in \mathcal{H}$ and does not impose the additional assumption (3). As for the least squares regression (Caponnetto and De Vito, 2007), such a regularity condition is expected to lead to faster rates. However, different from the least squares case which relies on the closed-form solution of the regularized estimator, taking into account the assumption (3) requires entirely different technical tricks.

Throughout the paper, $C$ denotes a generic positive constant. The following assumptions are used.

(A1) $s_j \le C j^{-\alpha}$ for some $\alpha > 1$. The smoothness assumption (3) holds.

(A2) We have the entropy condition $\log N(\varepsilon, \mathcal{H}(1), L_\infty) \le (C/\varepsilon)^{2/\alpha}$, where $\mathcal{H}(1) = \{f \in \mathcal{H} : \|f\|_{\mathcal{H}} \le 1\}$.

(A3) (Sup-norm assumption) $\|f\|_\infty \le C\|f\|^{1-\frac{1}{\alpha}}\|f\|_{\mathcal{H}}^{\frac{1}{\alpha}}, \forall f \in \mathcal{H}$.

(A4) The conditional density of $e = y - f_0(x)$ given $x$, denoted by $\gamma(.|x)$, and its derivative are uniformly bounded. $\gamma(0|x)$ is bounded away from zero uniformly for $x \in [0, 1]$.

The assumption (A1) on the polynomial decay of eigenvalues is commonly assumed in the literature. The entropy condition (A2) is stronger than assuming $s_j \le C j^{-\alpha}$. Such an assumption is not uncommon and used in Müller and van de Geer (2015). Theorem 6.26 of Steinwart and Christmann (2008) showed that it is satisfied if the RKHS has a kernel with derivatives of sufficiently high orders. The sup-norm assumption is also used in Steinwart et al. (2009); Suzuki and Sugiyama (2013), which yields a better control for the sup-norm of a function in the RKHS. The sup-norm assumption is satisfied if the RKHS is a Sobolev space or is continuously embeddable in a Sobolev space.

Assumptions (A1)-(A4) are required for the proof for the distributed estimator. For the central estimator, we can remove (A2) and (A3) and add the following assumption.

(A2') For some sufficiently small constant $u > 0$, $E[\rho_\tau(y - f(x))] - E[\rho_\tau(y - f_0(x))] \ge C\|f - f_0\|^2, \forall f : \|f - f_0\| \le u, \|f - f_0\|_{\mathcal{H}} \le 1$.

Assumption (A2') is a strong convexity condition of the population loss. It can be shown that (A2') is implied by (A3) and (A4). It also holds if the conditional density $\gamma(.|x)$ is uniformly bounded away from zero ((A4) only requires $\gamma(0|x)$ is uniformly bounded away from zero). The following result gives the bound for the central estimator.

**Theorem 1** *Assume (A1), (A2') and (A4) hold. Setting $\lambda \asymp N^{-\frac{\alpha}{(2r+1)\alpha+1}}$, we have $\|\widehat{f} - f_0\| = O_p(N^{-\frac{(2r+1)\alpha}{2(2r+1)\alpha+2}})$.*

We first state a lemma on the relevant empirical process. The proof relies on the Rademacher complexity $E[\sup_{\|f\| \le u, \|f\|_{\mathcal{H}} \le 1} \frac{1}{N}\sum_{i=1}^N \sigma_i f(x_i)]$, where $\sigma_i \in \{-1, 1\}$ are i.i.d. Rademacher variables. The Rademacher complexity is a characterization of the complexity of the function space. As shown in Mendelson (2002), we have a bound

$$E[\sup_{\|f\| \le u, \|f\|_{\mathcal{H}} \le 1} \frac{1}{N}\sum_{i=1}^N \sigma_i f(x_i)] \le C\mathcal{R}^*(u), \tag{4}$$

where $\mathcal{R}^*(u) = \sqrt{\frac{1}{N}\sum_{j=1}^\infty \min\{s_j, u^2\}}$. When $s_j \le C j^{-\alpha}$, it is easy to see $\mathcal{R}^*(u) \le C\mathcal{R}(u)$ with $\mathcal{R}(u) := \frac{u^{1-\frac{1}{\alpha}}}{\sqrt{N}}$.

**Lemma 1** *For any $u$ with $\mathcal{R}(u) \leq u^2$, with probability at least $1 - e^{-CN\mathcal{R}^2(u)/u^2}$,*

$$\left| \frac{1}{N} \sum_i (\rho_\tau(y_i - f(x_i)) - \rho_\tau(y_i - f_0(x_i))) - E[\rho_\tau(y - f(x)) - \rho_\tau(y - f_0(x))] \right|$$

$$\leq \quad C\frac{\mathcal{R}(u)}{u}\|f - f_0\| + C\mathcal{R}(u)\|f - f_0\|_{\mathcal{H}}, \forall f \in \mathcal{H}.$$

**Sketches of Proof of Theorem 1.** We prove by way of contradiction. Assuming $\|\widehat{f} - f_0\|$ is large, we try to establish

$$\frac{1}{N} \sum_i \rho_\tau(y_i - f(x_i)) + \lambda\|f\|_{\mathcal{H}}^2 > \frac{1}{N} \sum_i \rho_\tau(y_i - f_0(x_i)) + \lambda\|f_0\|_{\mathcal{H}}^2,$$

which will contradict the fact that $\widehat{f}$ is the minimizer of the penalized loss. The difference of the quantile loss on the two sides is approximated by its population version with the help of Lemma 1 and bounded below by assumption (A2'), while the difference of the penalties are carefully bounded by a nontrivial application of Young's inequality for operators, taking into account the source condition.

**Proof of Theorem 1.** In the proof we choose $u$ that satisfies $\mathcal{R}(u) = u^{2+2r}$ (that is, $u = N^{-\frac{\alpha}{2\alpha(2r+1)+2}}$, since $\mathcal{R}(u) = u^{1-\frac{1}{\alpha}}/\sqrt{N}$).

By way of contradiction, suppose $\|\widehat{f} - f_0\|^2 \geq Lu^{2(1+2r)}$ for some sufficiently large constant $L$. This means

$$\inf_{\|f-f_0\|^2 \geq Lu^{2(1+2r)}} \frac{1}{N} \sum_i \rho_\tau(y_i - f(x_i)) + \lambda\|f\|_{\mathcal{H}}^2 - \frac{1}{N} \sum_i \rho_\tau(y_i - f_0(x_i)) - \lambda\|f_0\|_{\mathcal{H}}^2 < 0.$$

By the convexity of the functional in $f$ in the above, we have

$$\inf_{\|f-f_0\|^2 = Lu^{2(1+2r)}} \frac{1}{N} \sum_i \rho_\tau(y_i - f(x_i)) + \lambda\|f\|_{\mathcal{H}}^2 - \frac{1}{N} \sum_i \rho_\tau(y_i - f_0(x_i)) - \lambda\|f_0\|_{\mathcal{H}}^2 < 0. \quad (5)$$

On the other hand, by assumption (A2') and Lemma 1, with probability at least $1 - e^{-CNu^{2+4r}}$ (note $Nu^{2+4r} = N^{\frac{1}{(1+2r)\alpha+1}} \to \infty$),

$$\begin{aligned}
& C\|f - f_0\|^2 \\
\leq \quad & E[\rho_\tau(y - f(x))] - E[\rho_\tau(y - f_0(x))] \\
\leq \quad & \lambda\|f_0\|_{\mathcal{H}}^2 - \lambda\|f\|_{\mathcal{H}}^2 + Cu^{-1}\mathcal{R}(u)\|f - f_0\| + C\mathcal{R}(u)\|f - f_0\|_{\mathcal{H}} \\
= \quad & -2\lambda\langle f_0, f - f_0\rangle_{\mathcal{H}} - \lambda\|f - f_0\|_{\mathcal{H}}^2 + Cu^{-1}\mathcal{R}(u)\|f - f_0\| + C\mathcal{R}(u)\|f - f_0\|_{\mathcal{H}} \\
\leq \quad & C\lambda^{\frac{1+2r}{2}}(\|f - f_0\| + \lambda^{1/2}\|f - f_0\|_{\mathcal{H}}) - \lambda\|f - f_0\|_{\mathcal{H}}^2 \\
& +Cu^{-1}\mathcal{R}(u)\|f - f_0\| + C\mathcal{R}(u)\|f - f_0\|_{\mathcal{H}} \\
\leq \quad & C\lambda^{\frac{1+2r}{2}}\|f - f_0\| + C\lambda^{1+2r} + \frac{\lambda}{4}\|f - f_0\|_{\mathcal{H}}^2 - \lambda\|f - f_0\|_{\mathcal{H}}^2 \\
& +Cu^{-1}\mathcal{R}(u)\|f - f_0\| + C\frac{\mathcal{R}^2(u)}{\lambda} + \frac{\lambda}{4}\|f - f_0\|_{\mathcal{H}}^2 \\
\leq \quad & C(\frac{\mathcal{R}(u)}{u} + \lambda^{\frac{1+2r}{2}})\|f - f_0\| + C\lambda^{1+2r} + C\frac{\mathcal{R}^2(u)}{\lambda} \\
= \quad & Cu^{1+2r}\|f - f_0\| + Cu^{2+4r}. \quad (6)
\end{aligned}$$

In the 5th line above, we used that

$$\begin{aligned}
& |\lambda\langle f_0, f - f_0\rangle_{\mathcal{H}}| \\
= \quad & \lambda|\langle \mathcal{L}^r g, f - f_0\rangle_{\mathcal{H}}| \\
= \quad & \lambda|\langle g, \mathcal{L}^r(f - f_0)\rangle_{\mathcal{H}}| \\
\leq \quad & C\lambda^{\frac{1}{2}+r}\|\lambda^{\frac{1}{2}-r}\mathcal{L}^r(f - f_0)\|_{\mathcal{H}} \\
\leq \quad & C\lambda^{\frac{1}{2}+r}\sqrt{\langle f - f_0, ((1-2r)\lambda + 2r\mathcal{L})(f - f_0)\rangle_{\mathcal{H}}} \\
\leq \quad & C\lambda^{\frac{1}{2}+r}\|f - f_0\| + \lambda^{1+r}\|f - f_0\|_{\mathcal{H}},
\end{aligned}$$

where the first line used assumption (3), and the second to last line used Young's inequality for positive operators $\lambda^{1-2r}\mathcal{L}^{2r} \leq (1-2r)\lambda + 2r\mathcal{L}$. (6) implies $\|f - f_0\| \leq Cu^{1+2r}$, contradicting $\|f - f_0\| = Lu^{1+2r}$ for $L$ sufficiently large, and the proof is complete. $\square$

The rate obtained for the central estimator is the same as the rate obtained for kernel ridge regression in Caponnetto and De Vito (2007); Lin et al. (2017) and thus cannot be improved in general. However, the proof of Theorem 1 does not yield clearly any bias and variance terms separately. As mentioned in the introduction, separate control of bias and variance is usually the key step in obtaining the learning rate of the distributed estimator. The empirical process of Lemma 1 cannot be utilized to get the approximate bias and variance terms, although such type of empirical process was popularly used in the literature (Belloni and Chernozhukov, 2011; Li et al., 2007; Lv et al., 2018). To deal with this problem, for the distributed estimator, we use the empirical process techniques not on the difference of the quantile loss as in Lemma 1, but instead on a higher-order difference taking into account the gradient of the loss also, resulting in what we will call the second-order empirical process. This will allow us to approximate the quantile loss by a weighted least squares loss, with results on distributed learning for the latter available from the literature (after minor modifications).

More specifically, we define $g(x, y, f) = \rho_\tau(y - f(x)) - \rho_\tau(y - f_0(x)) + (\tau - I\{y - f_0(x) \leq 0\})(f(x) - f_0(x))$. If the loss were differentiable, $g$ would be the residual in the first-order Taylor's expansion of the loss. For notational simplicity, we denote $\epsilon = \tau - I\{e \leq 0\}$ with $e = y - f_0(x)$. The key lemma that controls the difference between the quantile estimator and the weighted least squares estimator is the following result involving second-order empirical processes. Due to the use of an additional term $(\tau - I\{y - f_0(x) \leq 0\})(f(x) - f_0(x))$ representing the first-order expansion in $g$, the bound in Lemma 2 is typically much smaller than that in Lemma 1 for risk difference.

**Lemma 2** *For any $u \in (0, 1)$, with probability $1 - \exp\{u^{-2s/\alpha}\}$ where $s = \frac{3}{2} - \frac{1}{2\alpha} > 1$,*

$$\sup_{\|f - f_0\| \leq u, \|f - f_0\|_{\mathcal{H}} \leq 1} (P_n - P)g(x, y, f) \leq C\left(\frac{u^s}{\sqrt{n}}\left(\frac{1}{u^s}\right)^{1/\alpha} + \frac{u^{1-1/\alpha}}{n}\left(\frac{1}{u^s}\right)^{2/\alpha}\right).$$

Based on Lemma 2, we can prove our main result for the distributed estimator. It is shown that under suitable conditions, the learning rate for the distributed estimator matches that of the central estimator.

**Theorem 2** *Under assumptions (A1)-(A4), setting $\lambda \asymp N^{-\frac{\alpha}{(2r+1)\alpha+1}}$, when $m \lesssim N^{\min\left\{\frac{\alpha^2(2r+1)-(8r+2)\alpha+2r+1}{2\alpha^2(2r+1)+2\alpha}, \frac{2\alpha r}{\alpha(2r+1)+1}\right\}}$, we have $\|\bar{f} - f_0\| = O_p(N^{-\frac{\alpha(2r+1)}{2\alpha(2r+1)+2}})$.*

**Remark 1** *We note that, unfortunately, the exponent here is quite messy, and the first term in $\min\left\{\frac{\alpha^2(2r+1)-(8r+2)\alpha+2r+1}{2\alpha^2(2r+1)+2\alpha}, \frac{2\alpha r}{\alpha(2r+1)+1}\right\}$ can be either bigger or smaller than the second term (since the explicit expression on when the first term is bigger is quite messy we do not choose to write it down). Also the expression $\alpha^2(2r+1) - (8r+2)\alpha + 2r + 1$ may not even be positive (if it is not positive, the bound becomes void). It is positive if $\alpha \geq 3$.*

**Sketches of Proof of Theorem 2.** From Lemma 2, we can show that $\widehat{f}_j$ which minimizes the regularized empirical loss can be well-approximated by $\widetilde{f}_j$ that minimizes the sum of three terms: $P_n((f(x) - f_0(x))\epsilon)$ (the first-order term in the definition of $g$), the Tikhorov regularization term, and the second-order term from Taylor's expansion of the *expected* loss. By definition $\widetilde{f}_j$ is a weighted kernel ridge estimator and the statistical bound for its average over $m$ machines can be more easily established. However, rigorous establishment of the closeness of $\widehat{f}_j$ and $\widetilde{f}_j$ requires more detailed analysis involving multiple applications of Lemma 2.

**Proof of Theorem 2.** By Knight's identity that $\rho_\tau(x-y) - \rho_\tau(x) = -y(\tau - I\{x \leq 0\}) + \int_0^y (I\{x \leq t\} - I\{x \leq 0\})dt$, we have

$$
\begin{aligned}
Pg(x, y, f) &= E\rho_\tau(y - f(x)) - E\rho_\tau(y - f_0(x)) = E\left[\int_0^{f(x)-f_0(x)} F(t|x) - F(0|x)dt\right] \\
&= \frac{1}{2}E[\gamma(0|x)(f(x) - f_0(x))^2] + O_p(E[(f(x) - f_0(x))^3]) \\
&= \frac{1}{2}E[\gamma(0|x)(f(x) - f_0(x))^2] + O_p(u^{3-1/\alpha}),
\end{aligned}
$$
(7)

using $E[(f(x) - f_0(x))^3] \leq \|f - f_0\|_\infty E[(f(x) - f_0(x))^2]$ and the sup-norm assumption, where $F(.|x)$ is the conditional cdf of $e = y - f_0(x)$.

Let $\widetilde{f}_j$ be the minimizer of $\frac{1}{2}E[\gamma(0|x)(f(x) - f_0(x))^2] + P_n((f(x) - f_0(x))\epsilon) + \lambda\|f\|_\mathcal{H}^2$. This is basically the least squares problem when the true model is given by $y = f_0(x) + \epsilon$, with two differences. One is that the loss is weighted by $\gamma(0|x)$ and the other is that the first term is the population counterpart of $\sum_i(r(0|x_i)(f(x_i) - f_0(x_i)))^2/n$. It is easy to see that $\widetilde{f}_j$ has a closed form $\widetilde{f}_j = (\mathcal{L}_\gamma + \lambda)^{-1}\mathcal{L}_\gamma f_0 + (\mathcal{L}_\gamma + \lambda)^{-1}\frac{\sum_{i=1}^n \epsilon_i K(x_i,.)}{n}$. If $E[\gamma(0|x)(f(x) - f_0(x))^2]$ is replaced by $\sum_i(r(0|x_i)(f(x_i) - f_0(x_i)))^2/n$, the minimizer would be $\widetilde{f}_j^* = (\mathcal{L}_{n\gamma} + \lambda)^{-1}\mathcal{L}_{n\gamma}f_0 + (\mathcal{L}_{n\gamma} + \lambda)^{-1}\frac{\sum_{i=1}^n \epsilon_i K(x_i,.)}{n}$, where $\mathcal{L}_{n\gamma}$, defined as $\mathcal{L}_{n\gamma}f = \frac{1}{n}\sum_{i=1}^n f(x_i)K(x_i,.)$, is the empirical version of $\mathcal{L}_\gamma$. We can see analysis of $\widetilde{f}_j$ would be similar to (and actually simpler than) the analysis of $\widetilde{f}_j^*$, and the latter was basically the problem studied in Lin et al. (2017), since we assumed $\gamma(0|x)$ is bounded away from zero and infinity and it makes no difference in statistical analysis. In particular, by Corollary 11 of Lin et al. (2017), we can get $\|\frac{1}{m}\sum_{j=1}^m \widetilde{f}_j - f_0\| = O_p(N^{-\frac{\alpha(2r+1)}{2\alpha(2r+1)+2}})$, if $m \leq N^{\frac{2\alpha r}{\alpha(2r+1)+1}}$. Furthermore, by combining Corollary 8 and Lemma 21 of Lin et al. (2017), we also get the local estimates satisfy $\|\widetilde{f}_j - f_0\| = O_p(\frac{N^{\frac{1}{2[(2r+1)\alpha+1]}}}{\sqrt{n}})$.

Now we bound $\|\widehat{f}_j - \widetilde{f}_j\|$. For simplicity of notation, define the function $h(u) = u^{3-1/\alpha} + \frac{u^s}{\sqrt{n}}\left(\frac{1}{u^s}\right)^{1/\alpha} + \frac{u^{1-1/\alpha}}{n}\left(\frac{1}{u^s}\right)^{2/\alpha}$.

By Lemma 2 and (7), for any $f \in \mathcal{H}$ with $\|f - f_0\| \leq Cu$ and $\|f - f_0\|_\mathcal{H} \leq 1$,

$$\frac{1}{n}\sum_i \rho_\tau(y_i - f(x_i)) - \frac{1}{n}\sum_i \rho_\tau(y_i - f_0(x_i)) + \frac{1}{n}\sum_i(\tau - I\{e_i \leq 0\})(f(x_i) - f_0(x_i))$$
$$-\frac{1}{2}P\{\gamma(0|x)(f(x) - f_0(x))^2\}$$
$$\leq Ch(u).$$

Let $u_1 = \frac{N^{\frac{1}{2[(2r+1)\alpha+1]}}}{\sqrt{n}}$, then

$$\frac{1}{n}\sum_i \rho_\tau(y_i - \widetilde{f}_j(x)) - \frac{1}{n}\sum_i \rho_\tau(y_i - f_0(x)) + \frac{1}{n}\sum_i(\tau - I\{e_i \leq 0\})(\widetilde{f}_j(x_i) - f_0(x_i))$$
$$-\frac{1}{2}P\{\gamma(0|x)(\widetilde{f}_j(x) - f_0(x))^2\}$$
$$\leq Ch(u_1).$$

Taking the difference of the two displayed above,

$$-\frac{1}{n}\sum_i \rho_\tau(y_i - f(x_i)) - \lambda\|f\|_\mathcal{H}^2 + \frac{1}{n}\sum_i \rho_\tau(y_i - \widetilde{f}_j(x_i)) + \lambda\|\widetilde{f}_j\|_\mathcal{H}^2$$
$$-\frac{1}{n}\sum_i(\tau - I\{e_i \leq 0\})(f(x_i) - \widetilde{f}_j(x_i))$$
$$+\frac{1}{2}P\{\gamma(0|x)(f(x) - f_0(x))^2\} - \frac{1}{2}P\{\gamma(0|x)(\widetilde{f}_j(x) - f_0(x))^2\} + \lambda\|f\|_\mathcal{H}^2 - \lambda\|\widetilde{f}_j\|_\mathcal{H}^2$$
$$\leq C(h(u) + h(u_1)). \tag{8}$$

Since $\widetilde{f}_j$ minimizes $\frac{1}{2}P\{\gamma(0|x)(f(x) - f_0(x))^2\} - \frac{1}{n}\sum_i \epsilon_i(f(x_i) - f_0(x_i)) + \lambda\|f\|_\mathcal{H}^2$, the first order optimality condition for $\widetilde{f}_j$ yields

$$-\frac{1}{n}\sum_{i=1}^n \epsilon_i K(x_i,.) + E[\gamma(0|x)(\widetilde{f}_j(x) - f_0(x))K(x,.)] + 2\lambda\widetilde{f}_j = 0. \tag{9}$$

Thus we have

$$\frac{1}{2}P\{\gamma(0|x)(f(x)-f_0(x))^2\} - \frac{1}{2}P\{\gamma(0|x)(\widetilde{f}_j(x)-f_0(x))^2\}$$

$$-\frac{1}{n}\sum_i \epsilon_i(f(x_i)-\widetilde{f}_j(x_i))$$

$$= \frac{1}{2}E[\gamma(0|x)(f(x)-\widetilde{f}_j(x))^2] + E[\gamma(0|x)(f(x)-\widetilde{f}_j(x))(\widetilde{f}_j(x)-f_0(x))]$$

$$-\frac{1}{n}\sum_i \epsilon_i(f(x_i)-\widetilde{f}_j(x_i))$$

$$= \frac{1}{2}E[\gamma(0|x)(f_j(x)-\widetilde{f}_j(x))^2] + \langle f-\widetilde{f}_j, E[\gamma(0|x)(\widetilde{f}_j(x)-f_0(x))K(x,.)]\rangle_{\mathcal{H}}$$

$$-\langle f-\widetilde{f}_j, \frac{1}{n}\sum_i \epsilon_i K(x_i,.)\rangle_{\mathcal{H}}$$

$$= \frac{1}{2}E[\gamma(0|x)(f(x)-\widetilde{f}_j(x))^2] - 2\lambda\langle \widetilde{f}_j, f-\widetilde{f}_j\rangle_{\mathcal{H}},$$

where the last step used (9). Using this in (8), we get

$$-\frac{1}{n}\sum_i \rho_\tau(y_i-f(x_i)) - \lambda\|f\|_{\mathcal{H}}^2 + \frac{1}{n}\sum_i \rho_\tau(y_i-\widetilde{f}_j(x_i)) + \lambda\|\widetilde{f}_j\|_{\mathcal{H}}^2$$

$$+\frac{1}{2}E[\gamma(0|x)(f(x)-\widetilde{f}_j(x))^2] - 2\lambda\langle \widetilde{f}_j, f-\widetilde{f}_j\rangle_{\mathcal{H}} + \lambda\|f\|_{\mathcal{H}}^2 - \lambda\|\widetilde{f}_j\|_{\mathcal{H}}^2$$

$$= -\frac{1}{n}\sum_i \rho_\tau(y_i-f(x_i)) - \lambda\|f\|_{\mathcal{H}}^2 + \frac{1}{n}\sum_i \rho_\tau(y_i-\widetilde{f}_j(x_i)) + \lambda\|\widetilde{f}_j\|_{\mathcal{H}}^2$$

$$+\frac{1}{2}E[\gamma(0|x)(f(x)-\widetilde{f}_j(x))^2] + \lambda\|f-\widetilde{f}_j\|_{\mathcal{H}}^2$$

$$\leq \quad C(h(u)+h(u_1)). \tag{10}$$

Let $u_2 = n^{-\frac{\alpha^2}{\alpha^2+4\alpha-1}}$. Note that this $u_2$ is the value of $u$ that makes $u^2 \asymp h(u)$. Let $u_3 = \sqrt{h(u_1)+h(u_2)}$. Suppose $\frac{1}{2}E[\gamma(0|x)(\widehat{f}_j(x)-\widetilde{f}_j(x))^2] + \lambda\|\widehat{f}_j-\widetilde{f}_j\|_{\mathcal{H}}^2 \geq Lu_3^2$ for some $L$ sufficiently large. By (10), and using the same arguments as in the proof of Theorem 1 that lead to (5), there exists some $f \in \mathcal{H}$ with $\frac{1}{2}E[\gamma(0|x)(f(x)-\widetilde{f}_j(x))^2] + \lambda\|f-\widetilde{f}_j\|_{\mathcal{H}}^2 = Lu_3^2$, and also satisfying $\frac{1}{n}\sum_i \rho_\tau(y_i-f(x_i)) + \lambda\|f\|_{\mathcal{H}}^2 \leq \frac{1}{n}\sum_i \rho_\tau(y_i-\widetilde{f}_j(x_i)) + \lambda\|\widetilde{f}_j\|_{\mathcal{H}}^2$, such that

$$\frac{1}{2}E[\gamma(0|x)(f(x)-\widetilde{f}_j(x))^2] + \lambda\|f-\widetilde{f}_j\|_{\mathcal{H}}^2 \leq C(h(\sqrt{L}u_3)+h(u_1)).$$

Because the left hand side above is $Lu_3^2$, this would lead to contradiction if $L$ is large enough. To see this, we note that $Lu_3^2 > Ch(\sqrt{L}u_3)$ since $u_3 > \sqrt{h(u_2)} = u_2$, and also $Lu_3^2 > Ch(u_1)$ since $u_3^2 = h(u_1)+h(u_2) > h(u_1)$.

Based on the previous arguments, we then have $\|\widehat{f}_j-\widetilde{f}_j\| \leq Cu_3$. Finally, the distributed estimator satisfies

$$\|\bar{f}-f_0\|$$

$$\leq \quad \|\frac{\sum_j \widehat{f}_j}{m} - \frac{\sum_j \widetilde{f}_j}{m}\| + \|\frac{\sum_j \widetilde{f}_j}{m} - f_0\|$$

$$\leq \quad \max_j \|\widehat{f}_j-\widetilde{f}_j\| + \|\frac{\sum_j \widetilde{f}_j}{m} - f_0\|.$$

As mentioned before, when $m \leq N^{\frac{2\alpha r}{\alpha(2r+1)+1}}$, the second term above has the optimal rate $\|\frac{1}{m}\sum_{j=1}^m \widetilde{f}_j - f_0\| = O_p(N^{-\frac{\alpha(2r+1)}{2\alpha(2r+1)+2}})$. To obtain the same rate for $\|\bar{f}-f_0\|$, we would require

$$n^{-\frac{\alpha^2}{\alpha^2+4\alpha-1}} \leq N^{-\frac{\alpha(2r+1)}{2\alpha(2r+1)+2}}. \tag{11}$$

Using $n = N/m$, (11) is equivalent to saying $m \leq N^{\frac{\alpha^2(2r+1)-(8r+2)\alpha+2r+1}{2\alpha^2(2r+1)+2\alpha}}$. In summary, if

$$m \leq N^{\min\left\{\frac{\alpha^2(2r+1)-(8r+2)\alpha+2r+1}{2\alpha^2(2r+1)+2\alpha}, \frac{2\alpha r}{\alpha(2r+1)+1}\right\}},$$

the optimal rate is achieved. $\qquad\qquad\qquad\qquad\qquad\qquad\qquad\qquad\qquad\qquad\qquad\qquad\square$

**Remark 2** *As stated in the theorem, for $\bar{f}$ to achieve the same rate as the non-distributed estimator, we need the constraint $m \lesssim N^{\min\left\{\frac{\alpha^2(2r+1)-(8r+2)\alpha+2r+1}{2\alpha^2(2r+1)+2\alpha}, \frac{2\alpha r}{\alpha(2r+1)+1}\right\}}$. On the other hand, from the proof, we have the convergence rate $N^{-\frac{\alpha(1+2r)}{2\alpha(1+2r)+2}} + (N/m)^{-\frac{\alpha^2}{\alpha^2+4\alpha+1}}$ even if the bound for $m$ is not satisfied (when the bound is not satisfied, we just have a non-optimal rate). In practice, of course it is hard to check whether a specific pair $(m, N)$ satisfies the bound since it is asymptotic in nature. On the other hand, in the setting that data are physically located on $m$ machines and one is unwilling to share data across different machines, then we have no other option and just accept this given $m$. In this situation, we do not have to be concerned with the choice of $m$.*

## 3    Some numerical illustrations

Here we illustrate the distributed quantile regression in relatively simple simulations. For a value $\tau \in (0,1)$, the sample is generated from the model $y_i = f_0(x_i) + (1+x_i)\sigma(\epsilon_i - \Phi^{-1}(\tau))$, where $x_i$ are generated uniformly on $[0,1]$, $\epsilon_i \sim N(0,1)$, $\Phi^{-1}$ is the quantile function of the standard normal distribution so that $f(x_i)$ is indeed the conditional $\tau$-quantile of $y_i$. We set $f_0(x) = \sin(2\pi x)$ and $\sigma = 0.5$. The simulations are carried out for different combinations of $n \in \{32, 64, 128, 256, 512, 1024\}$ and $m \in \{1, 2, 4, 8, 16, 32\}$, with the total sample size of $N = nm$. Note that $m = 1$ actually corresponds to the central estimator. For the RKHS, we use the Sobolev space of the second order. The tuning parameter $\lambda$ is chosen to minimize the errors on independently generated test data. In each setting, 100 repetitions are used to compute the average squared estimation error $\|\bar{f} - f_0\|^2$.

Choice of $\lambda$ is an important practical matter. Theorem 2 shows the choice of $\lambda \sim N^{-\frac{\alpha}{(2r+1)\alpha+1}}$ depends on the total sample size $N$, which indicates that tuning parameter chosen based on local data is not optimal. In our simulations, we use independently generated data to select the optimal $\lambda$ that ensures $\bar{f}$ ($\bar{f}$ implicitly depends on $\lambda$) is the one that achieves the smallest prediction error on the test data. When independently generated data is not available, either hold-out data or global $K$-fold cross-validation can be used, which requires coordination of the central machine. If the hold-out data is maintained by the central machine, the central machine can evaluate and compare different values of $\lambda$. If $K$-fold cross-validation is carried out on the local machines, the central machine needs to send $\bar{f}$ back to the local machines so that the local machines can calculate the cross-validation errors, and then send the errors to the central machine. This needs to be done $K$ times for $K$-fold CV. Xu et al. (2018, 2019) developed a generalized approximate cross-validation criterion in the least-squares case and established its asymptotic optimality. We expect that similar criterion developed for quantile regression (Yuan, 2006) can be adapted in our case. However, even in the non-distributed setting, Yuan (2006) did not establish the asymptotic properties of the criterion, mainly investigating its performance from a computational viewpoint. Thus it remains an open question whether such a criterion has a desirable asymptotic property. The investigation of such problems requires additional study by itself and thus we do not use this in the current work.

The simulation results are shown in Table 1 for $\tau = 0.5$. We can examine the table in several ways. First, each row of the table indicates what happens when one fixes the number of observations on each machine while increasing the number of machines (the total sample size of course also increases). In this case, we see the errors decrease with the number of machines since the total sample size increases. Second, each column of the table indicates what happens if the number of machines are fixed while the number of observations increases proportionally with the total sample size. Here we also see the error decreases. Third, the results along the anti-diagonal directions shows the results when the total sample size $N$ is fixed while we change the number of machines $m$. For example, the numbers shaded in yellow all corresponds to $N = 512$ while the numbers shaded in blue corresponds to $N = 1024$. Here we see that the error in general increases with the number of machines (up to some random fluctuations). This is natural since the central estimator should perform best if it is feasible to compute. Finally, we observe that the table shows the divide-and-conquer strategy is useful when $N$ is large. For example, in the table, the smallest error is achieved when $n = 1024, m = 32$,

Table 1: The estimation errors for different pairs of $(n, m)$ when $\tau = 0.5$. The total sample size is $N = nm$.

| $n$ \ $m$ | 1 | 2 | 4 | 8 | 16 | 32 |
|---|---|---|---|---|---|---|
| 32 | 2.765 | 1.211 | 0.898 | 0.466 | 0.316 | 0.221 |
| 64 | 1.226 | 0.653 | 0.371 | 0.221 | 0.108 | 0.062 |
| 128 | 0.616 | 0.386 | 0.229 | 0.099 | 0.053 | 0.023 |
| 256 | 0.346 | 0.199 | 0.105 | 0.058 | 0.029 | 0.013 |
| 512 | 0.186 | 0.086 | 0.046 | 0.020 | 0.011 | 0.008 |
| 1024 | 0.089 | 0.047 | 0.020 | 0.010 | 0.008 | 0.004 |

corresponding to $N = 32768$. It would be hard to deal with such a large sample size in the standard implementation (the kernel method is always computationally burdensome) without using some advanced techniques to approximate the solution, if such techniques exist at all.

The simulation results for other values of $\tau$ are presented in Appendix A. We also perform an additional simulation with 3-dimensional predictors in Appendix B.

## 4   Conclusion

In this paper, we established the convergence rate of the nonparametric quantile estimator in the distributed learning setting under the RKHS framework. The second-order empirical process allows us to approximate the estimator by the estimator obtained from a weighted least squares problem, which in turn has an explicit bias and variance decomposition as done in the existing literature. However, the theoretical drawback here is that this strategy currently does not allow us to obtain the optimal rate for the entire range $\alpha \in (1, \infty)$ (we require $\alpha$ to be sufficiently large). It may be the case that some other entirely different proof strategy is required to get a more elegant result.

In practice, it is always hard to estimate extreme conditional quantiles with $\tau$ very close to 0 or 1. For any fixed $\tau$ in the open interval $(0, 1)$, we do not have theoretical problems since we always require $m$ to be much smaller than $N$ and thus typically the local sample size $n = N/M$ is still quite large in the big data setting. Theoretically, the statistical convergence rate inversely depends on the density (this is well-known in the parametric setting, although in the nonparametric setting we are not aware of explicit mentioning of this in the literature). At the tail of the distribution, the density is typically small and thus the statistical error is large, which can also be seen from our simulation results for $\tau = 0.1$ and 0.9 in the Supplementary Appendix. Extreme conditional quantile regression is a separate discipline of its own and has been studied, for example, in Chernozhukov and Du (2008); Wang et al. (2012); Wang and Li (2013). However, even in the non-distributed setting we are not aware of any theoretical study of this problem in the framework of RKHS and it remains unclear whether some techniques can be adapted to this setting.

## Acknowledgments and Disclosure of Funding

This project is partially supported by NSFC Project 11871411 at the Shenzhen Research Institute, City University of Hong Kong; and partially supported by the Hong Kong Research Grants Council (RGC) General Research Fund under Grant 11300519, 11300721, 11311822 and 11301423.

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
