# Appendix A: Additional numerical results for one-dimensional case

Here we report the simulation results under the setup of the main paper for $\tau \in \{0.1, 0.2, 0.3, 0.4, 0.6, 0.7, 0.8, 0.9\}$.

Table 2: The estimation errors for different pairs of $(n, m)$ when $\tau = 0.1$. The total sample size is $N = nm$.

| $n$ \ $m$ | 1 | 2 | 4 | 8 | 16 | 32 |
|---|---|---|---|---|---|---|
| 32 | 3.074 | 2.474 | 1.710 | 0.977 | 0.753 | 0.530 |
| 64 | 2.258 | 1.153 | 0.708 | 0.506 | 0.332 | 0.343 |
| 128 | 0.964 | 0.521 | 0.292 | 0.182 | 0.124 | 0.096 |
| 256 | 0.604 | 0.325 | 0.199 | 0.115 | 0.076 | 0.053 |
| 512 | 0.290 | 0.150 | 0.088 | 0.035 | 0.022 | 0.009 |
| 1024 | 0.167 | 0.091 | 0.038 | 0.022 | 0.013 | 0.006 |

Table 3: The estimation errors for different pairs of $(n, m)$ when $\tau = 0.2$. The total sample size is $N = nm$.

| $n$ \ $m$ | 1 | 2 | 4 | 8 | 16 | 32 |
|---|---|---|---|---|---|---|
| 32 | 2.861 | 2.009 | 1.157 | 0.638 | 0.489 | 0.343 |
| 64 | 1.382 | 0.735 | 0.402 | 0.221 | 0.139 | 0.093 |
| 128 | 0.788 | 0.369 | 0.215 | 0.116 | 0.065 | 0.036 |
| 256 | 0.447 | 0.257 | 0.153 | 0.091 | 0.063 | 0.042 |
| 512 | 0.247 | 0.128 | 0.073 | 0.045 | 0.029 | 0.019 |
| 1024 | 0.124 | 0.058 | 0.031 | 0.017 | 0.008 | 0.003 |

Table 4: The estimation errors for different pairs of $(n, m)$ when $\tau = 0.3$. The total sample size is $N = nm$.

| $n$ \ $m$ | 1 | 2 | 4 | 8 | 16 | 32 |
|---|---|---|---|---|---|---|
| 32 | 2.958 | 1.261 | 1.012 | 0.507 | 0.373 | 0.259 |
| 64 | 1.655 | 0.856 | 0.488 | 0.254 | 0.163 | 0.097 |
| 128 | 0.755 | 0.496 | 0.255 | 0.158 | 0.085 | 0.047 |
| 256 | 0.390 | 0.283 | 0.109 | 0.067 | 0.027 | 0.009 |
| 512 | 0.171 | 0.109 | 0.050 | 0.036 | 0.018 | 0.010 |
| 1024 | 0.113 | 0.052 | 0.029 | 0.017 | 0.009 | 0.004 |

Table 5: The estimation errors for different pairs of $(n, m)$ when $\tau = 0.4$. The total sample size is $N = nm$.

| $n$ \ $m$ | 1 | 2 | 4 | 8 | 16 | 32 |
|---|---|---|---|---|---|---|
| 32 | 3.469 | 1.371 | 0.717 | 0.450 | 0.193 | 0.149 |
| 64 | 1.803 | 0.782 | 0.381 | 0.193 | 0.108 | 0.055 |
| 128 | 0.826 | 0.451 | 0.208 | 0.131 | 0.083 | 0.049 |
| 256 | 0.369 | 0.170 | 0.083 | 0.050 | 0.022 | 0.010 |
| 512 | 0.168 | 0.096 | 0.052 | 0.024 | 0.013 | 0.006 |
| 1024 | 0.086 | 0.048 | 0.025 | 0.013 | 0.008 | 0.004 |

Table 6: The estimation errors for different pairs of $(n, m)$ when $\tau = 0.6$. The total sample size is $N = nm$.

| $n$ \ $m$ | 1 | 2 | 4 | 8 | 16 | 32 |
|---|---|---|---|---|---|---|
| 32 | 1.824 | 1.343 | 0.777 | 0.393 | 0.227 | 0.159 |
| 64 | 1.662 | 0.802 | 0.497 | 0.266 | 0.170 | 0.104 |
| 128 | 0.713 | 0.418 | 0.242 | 0.111 | 0.048 | 0.021 |
| 256 | 0.390 | 0.184 | 0.112 | 0.064 | 0.032 | 0.013 |
| 512 | 0.161 | 0.087 | 0.052 | 0.026 | 0.013 | 0.005 |
| 1024 | 0.117 | 0.060 | 0.027 | 0.016 | 0.007 | 0.003 |

Table 7: The estimation errors for different pairs of $(n, m)$ when $\tau = 0.7$. The total sample size is $N = nm$.

| $n$ \ $m$ | 1 | 2 | 4 | 8 | 16 | 32 |
|---|---|---|---|---|---|---|
| 32 | 2.947 | 1.593 | 0.785 | 0.407 | 0.231 | 0.158 |
| 64 | 1.443 | 0.761 | 0.373 | 0.195 | 0.151 | 0.102 |
| 128 | 0.727 | 0.385 | 0.199 | 0.121 | 0.063 | 0.028 |
| 256 | 0.390 | 0.183 | 0.119 | 0.069 | 0.032 | 0.019 |
| 512 | 0.216 | 0.119 | 0.063 | 0.033 | 0.018 | 0.007 |
| 1024 | 0.111 | 0.059 | 0.027 | 0.012 | 0.007 | 0.003 |

Table 8: The estimation errors for different pairs of $(n, m)$ when $\tau = 0.8$. The total sample size is $N = nm$.

| $n$ \ $m$ | 1 | 2 | 4 | 8 | 16 | 32 |
|---|---|---|---|---|---|---|
| 32 | 3.723 | 1.638 | 0.980 | 0.648 | 0.456 | 0.368 |
| 64 | 1.358 | 1.059 | 0.483 | 0.251 | 0.159 | 0.118 |
| 128 | 0.868 | 0.455 | 0.198 | 0.114 | 0.061 | 0.037 |
| 256 | 0.403 | 0.194 | 0.118 | 0.064 | 0.036 | 0.027 |
| 512 | 0.199 | 0.105 | 0.064 | 0.033 | 0.019 | 0.010 |
| 1024 | 0.101 | 0.055 | 0.031 | 0.018 | 0.008 | 0.004 |

Table 9: The estimation errors for different pairs of $(n, m)$ when $\tau = 0.9$. The total sample size is $N = nm$.

| $n$ \ $m$ | 1 | 2 | 4 | 8 | 16 | 32 |
|---|---|---|---|---|---|---|
| 32 | 4.086 | 2.479 | 1.871 | 1.348 | 1.115 | 1.064 |
| 64 | 1.624 | 1.043 | 0.726 | 0.396 | 0.315 | 0.258 |
| 128 | 1.557 | 0.699 | 0.404 | 0.219 | 0.166 | 0.120 |
| 256 | 0.482 | 0.218 | 0.137 | 0.089 | 0.045 | 0.021 |
| 512 | 0.284 | 0.152 | 0.075 | 0.056 | 0.022 | 0.012 |
| 1024 | 0.178 | 0.078 | 0.042 | 0.023 | 0.012 | 0.005 |

## Appendix B: Numerical results for three-dimensional case

Here consider a simulation with three-dimensional predictor. The sample is generated from the model $y_i = f_0(x_i) + (1 + x_{i1}^2)\sigma(\epsilon_i - F^{-1}(\tau))$, where $x_i = (x_{i1}, x_{i2}, x_{i2})$ is uniformly generated from $[0,1]^3$, $f_0(x) = 4\exp\{-x_1^2 + x_2^2\} - 4x_3$, and here $F$ denotes a student's t-distribution with 4 degrees of freedom. Gaussian kernel is used with bandwidth specified as the median distance between two predictors. Other settings are the same as the one-dimensional case. The results for $\tau \in \{0.1, 0.3, 0.5, 0.7, 0.9\}$ are reported in Tables 10–14. Qualitatively, the interpretations of the results are the same as before, distributed learning using simple averaging works satisfactorily, with errors decreasing with the increase of either $m$ or $n$.

Table 10: For the 3-dimensional case, the estimation errors for different pairs of $(n, m)$ when $\tau = 0.1$. The total sample size is $N = nm$.

| $n$ \ $m$ | 1 | 2 | 4 | 8 | 16 | 32 |
|---|---|---|---|---|---|---|
| 32 | 4.975 | 3.784 | 2.871 | 2.004 | 1.511 | 1.281 |
| 64 | 4.624 | 2.944 | 1.450 | 1.129 | 0.754 | 0.597 |
| 128 | 2.315 | 1.651 | 1.087 | 0.513 | 0.352 | 0.215 |
| 256 | 1.389 | 0.774 | 0.390 | 0.184 | 0.079 | 0.043 |
| 512 | 0.616 | 0.419 | 0.181 | 0.090 | 0.044 | 0.019 |
| 1024 | 0.442 | 0.207 | 0.128 | 0.074 | 0.041 | 0.019 |

Table 11: For the 3-dimensional case, the estimation errors for different pairs of $(n, m)$ when $\tau = 0.3$. The total sample size is $N = nm$.

| $n$ \ $m$ | 1 | 2 | 4 | 8 | 16 | 32 |
|---|---|---|---|---|---|---|
| 32 | 3.914 | 2.073 | 1.539 | 0.803 | 0.437 | 0.274 |
| 64 | 2.444 | 0.994 | 0.780 | 0.388 | 0.226 | 0.165 |
| 128 | 0.825 | 0.589 | 0.283 | 0.154 | 0.078 | 0.032 |
| 256 | 0.535 | 0.265 | 0.167 | 0.102 | 0.051 | 0.029 |
| 512 | 0.310 | 0.169 | 0.079 | 0.057 | 0.026 | 0.013 |
| 1024 | 0.143 | 0.085 | 0.041 | 0.018 | 0.011 | 0.006 |

Table 12: For the 3-dimensional case, the estimation errors for different pairs of $(n, m)$ when $\tau = 0.5$. The total sample size is $N = nm$.

| $n$ \ $m$ | 1 | 2 | 4 | 8 | 16 | 32 |
|---|---|---|---|---|---|---|
| 32 | 3.451 | 2.637 | 1.702 | 1.255 | 0.780 | 0.665 |
| 64 | 1.934 | 0.889 | 0.566 | 0.267 | 0.139 | 0.064 |
| 128 | 0.974 | 0.582 | 0.380 | 0.173 | 0.116 | 0.072 |
| 256 | 0.550 | 0.361 | 0.157 | 0.081 | 0.058 | 0.032 |
| 512 | 0.293 | 0.134 | 0.083 | 0.035 | 0.018 | 0.008 |
| 1024 | 0.140 | 0.071 | 0.032 | 0.018 | 0.010 | 0.004 |

Table 13: For the 3-dimensional case, the estimation errors for different pairs of $(n, m)$ when $\tau = 0.7$. The total sample size is $N = nm$.

| $n$ \ $m$ | 1 | 2 | 4 | 8 | 16 | 32 |
|---|---|---|---|---|---|---|
| 32 | 3.440 | 2.547 | 1.310 | 0.762 | 0.605 | 0.501 |
| 64 | 3.128 | 1.329 | 0.842 | 0.384 | 0.167 | 0.084 |
| 128 | 1.129 | 0.616 | 0.369 | 0.147 | 0.084 | 0.034 |
| 256 | 0.702 | 0.363 | 0.195 | 0.105 | 0.070 | 0.031 |
| 512 | 0.379 | 0.192 | 0.110 | 0.049 | 0.024 | 0.009 |
| 1024 | 0.156 | 0.077 | 0.034 | 0.019 | 0.011 | 0.006 |

Table 14: For the 3-dimensional case, the estimation errors for different pairs of $(n, m)$ when $\tau = 0.9$. The total sample size is $N = nm$.

| $n$ \ $m$ | 1 | 2 | 4 | 8 | 16 | 32 |
|---|---|---|---|---|---|---|
| 32 | 5.208 | 5.104 | 3.099 | 2.217 | 1.415 | 1.253 |
| 64 | 3.367 | 2.324 | 1.322 | 0.770 | 0.423 | 0.257 |
| 128 | 2.656 | 1.370 | 0.830 | 0.399 | 0.278 | 0.215 |
| 256 | 1.056 | 0.706 | 0.329 | 0.226 | 0.134 | 0.091 |
| 512 | 0.686 | 0.376 | 0.209 | 0.106 | 0.059 | 0.027 |
| 1024 | 0.392 | 0.233 | 0.138 | 0.066 | 0.042 | 0.020 |

## Appendix C: Proof of Lemmas

**Proof of Lemma 1.** By the standard symmetrization argument (Pollard, 1984), we have

$$
E\left[\sup_{f \in \mathcal{H}} \frac{|\frac{1}{N}\sum_i (\rho_\tau(y_i - f(x_i)) - \rho_\tau(y_i - f_0(x_i))) - E[\rho_\tau(y - f(x)) - \rho_\tau(y - f_0(x))]|}{u^{-1}\|f - f_0\| + \|f - f_0\|_{\mathcal{H}}}\right]
$$

$$
= E\left[\sup_{f \in \mathcal{H}}\left|(P - P_N)\frac{\rho_\tau(y - f(x)) - \rho_\tau(y - f_0(x))}{u^{-1}\|f - f_0\| + \|f - f_0\|_{\mathcal{H}}}\right|\right]
$$

$$
\leq CE\left[\sup_{f \in \mathcal{H}}\left|\frac{\frac{1}{N}\sum_i \sigma_i (f - f_0)(x_i)}{u^{-1}\|f - f_0\| + \|f - f_0\|_{\mathcal{H}}}\right|\right]
$$

$$
\leq C\mathcal{R}(u), \tag{A.1}
$$

where the second to last inequality follows from the contraction inequality for the Rademacher complexity (see, e.g., Theorem 2.2 of Koltchinskii (2011)), and the last bound follows from (4).

For the left-hand side of (A.1), since (using the Lipschitz continuity of $\rho_\tau$)

$$
\left|\frac{\rho_\tau(y - f(x)) - \rho_\tau(y - f_0(x))}{u^{-1}\|f - f_0\| + \|f - f_0\|_{\mathcal{H}}}\right|
$$

$$
\leq C\left|\frac{f(x) - f_0(x)}{u^{-1}\|f - f_0\| + \|f - f_0\|_{\mathcal{H}}}\right|
$$

$$
\leq C\frac{\|f - f_0\|_\infty}{u^{-1}\|f - f_0\| + \|f - f_0\|_{\mathcal{H}}}
$$

$$
\leq C,
$$

and

$$
Var\left(\frac{\rho_\tau(y - f(x)) - \rho_\tau(y - f_0(x))}{u^{-1}\|f - f_0\| + \|f - f_0\|_{\mathcal{H}}}\right)
$$

$$
\leq CVar\left(\frac{f(x) - f_0(x)}{u^{-1}\|f - f_0\| + \|f - f_0\|_{\mathcal{H}}}\right)
$$

$$
\leq Cu^2,
$$

using the concentration inequality (see, e.g., the Bousquet bound in Chapter 2 of Koltchinskii (2011)),

$$\sup_{f \in \mathcal{H}} \frac{|\frac{1}{N} \sum_i (\rho_\tau(y_i - f(x_i)) - \rho_\tau(y_i - f_0(x_i))) - E[\rho_\tau(y - f(x)) - \rho_\tau(y - f_0(x))]|}{u^{-1}\|f - f_0\| + \|f - f_0\|_{\mathcal{H}}}$$

$$\leq CE\left[\sup_{f \in \mathcal{H}} \frac{|\frac{1}{N} \sum_i (\rho_\tau(y_i - f(x_i)) - \rho_\tau(y_i - f_0(x_i))) - E[\rho_\tau(y - f(x)) - \rho_\tau(y - f_0(x))]|}{u^{-1}\|f - f_0\| + \|f - f_0\|_{\mathcal{H}}}\right]$$

$$+ Cu\sqrt{t/n} + C(t/n),$$

with probability at least $1 - e^{-Ct}$. We can then set $t = N\mathcal{R}^2(u)/u^2$ to get

$$\sup_{f \in \mathcal{H}} \frac{|\frac{1}{N} \sum_i (\rho_\tau(y_i - f(x_i)) - \rho_\tau(y_i - f_0(x_i))) - E[\rho_\tau(y - f(x)) - \rho_\tau(y - f_0(x))]|}{u^{-1}\|f - f_0\| + \|f - f_0\|_{\mathcal{H}}}$$

$$\leq C\mathcal{R}(u), \tag{A.2}$$

with probability at least $1 - e^{-CN\mathcal{R}^2(u)/u^2}$, which finishes the proof. $\square$

**Proof of Lemma 2.** Define the class of functions $\mathcal{G} = \{g(x, y, f) = \rho_\tau(y - f(x)) - \rho_\tau(y - f_0(x)) + (\tau - I\{y - f_0(x) \leq 0\})(f(x) - f_0(x)) : \|f - f_0\| \leq u, \|f - f_0\|_{\mathcal{H}} \leq 1\}$. Obviously $|g(x, y, f_1) - g(x, y, f_2)| \leq C|f_1(x) - f_2(x)|$ and thus the covering number of $\mathcal{G}$ is bounded by

$$N(\epsilon, \mathcal{G}, L_\infty) \leq N(C\epsilon, \mathcal{H}(1), L_\infty) \leq \exp\{(C/\epsilon)^{2/\alpha}\}.$$

Suppose $\|f - f_0\| \leq u$ and $\|f - f_0\|_{\mathcal{H}} \leq 1$. We can also easily see that $|g(x, y, f)| \leq C|f(x) - f_0(x)| \cdot I\{|e| \leq |f(x) - f_0(x)|\}$. Thus we have $\|g(x, y, f)\| \leq Cu\|f - f_0\|_\infty^{1/2} \leq Cu^{\frac{3}{2} - \frac{1}{2\alpha}} = Cu^s$ (using the sup-norm assumption) and $\|g(x, y, f)\|_\infty \leq C\|f - f_0\|_\infty \leq Cu^{1-1/\alpha}$.

Using Theorem 3.12 of Koltchinskii (2011), which provides an upper bound of the Rademacher complexity in terms of the covering number, we get

$$E[\sup_{g \in \mathcal{G}}(P_n - P)g] \leq C\left(\frac{u^s}{\sqrt{n}}\left(\frac{1}{u^s}\right)^{1/\alpha} + \frac{u^{1-1/\alpha}}{n}\left(\frac{1}{u^s}\right)^{2/\alpha}\right).$$

Using Talagrand's concentration inequality, we have with probability $1 - e^{-t}$,

$$\sup_{g \in \mathcal{G}}(P_n - P)g \leq CE[\sup_{g \in \mathcal{G}}(P_n - P)g] + C\sqrt{\frac{t}{n}u^{2s}} + C\frac{tu^{1-\frac{1}{\alpha}}}{n},$$

and setting $t = u^{-2s/\alpha}$ proves the lemma. $\square$