# OpenReview forum: "Distributed Learning of Conditional Quantiles in the Reproducing Kernel Hilbert Space"
_NeurIPS.cc/2022/Conference — NeurIPS 2022 Accept_

### Official Review · Reviewer_oHGh · 2022-07-07

**Rating:** 4
**Confidence:** 1
**Soundness:** 2 fair
**Presentation:** 1 poor
**Contribution:** 2 fair

**Summary:**

The work focuses on analysis of distributed learning of nonparametric conditional quantiles with Tikhonov regularization in RKHS. A difficulty in this setting is decomposing the mean squared error into bias and variance terms; a necessity for establishing rates in the divide-and-conquer setting of distributed learning. The authors suggest combating this via approximating the quantile loss by a weighted square loss through a second-order empirical process where an explicit solution to the bias and variance terms is available. Furthermore, the authors suggest approximating the loss leads to approximating the estimator. Through this regime, learning rates for the central and distributed estimators are analyzed. The paper concludes with a simple numerical illustration demonstrating the efficacy of the divide-and-conquer method.

**Questions:**

I have no questions regarding the technical details of this work as it is not my area of expertise.

**Limitations:**

The work is theoretical and has no immediate potential negative societal impact. Limitations such as requiring $\alpha$ to be sufficiently large in their analysis were mentioned by the authors.

**Strengths And Weaknesses:**

The subject of the paper is not my expertise, so I will leave validation of the mathematics to the other reviewers. Instead, my main concern of this work is that the presentation of the results is rough and seemingly rushed. The paper is mathematically dense and thus rather difficult to follow. The bulk of the main paper is composed of theorems/lemmas followed by their proofs with little explanations in between results for clarifications. The problem tackled in this work appears to have some significance as a thorough study of the risk bounds for quantile risk in the RKHS with the distributed learning setting is to my understanding not well studied. Lastly, the numerical results illustrate the efficacy of distributed learning in this setting, but it appears to be tangential to the analysis of the learning rate which was the focus of the bulk of the work (although the authors have brought up this concern).

Minor: syntactic - in the inequality and equality below line 180 and 183 respectively, replace $(1/2)$ with $\frac{1}{2}$. Same issue with $(1/N)$ in the inequality below line 111, 112, 118.

---

> ### Author Response · Authors · 2022-07-31
> **Our point-by-point responses to your review**
>
> We are grateful to you for your comments and suggestions, and sincerely appreciate the time you spent in the review. Our point-by-point responses to your comments are given below.
>
> ### **Proof presentation**
> Thanks for your comments. We have tried to add more details in some parts of the proof, and gives more explanations between theorem and lemmas with a **proof sketch added for both theorems**. We hope this can help a little for ease of reading.   The proof of Lemmas is now moved to the supplementary appendix.
>
> ### **Practical considerations**
>
> The main contribution of the current work is the theoretical result, which established the distributed learning rate that under some settings is the same as the rate for the standard centralized estimator. In particular, Theorem 2 shows the choice of $\lambda\sim N^{-\frac{\alpha}{(2r+1)\alpha+1}}$ depends on the total sample size $N$, which indicates that tuning parameter chosen on local data is not optimal. In our simulations, we use independently generated data to select the optimal $\lambda$ that ensure $\bar f$ ($\bar f$ implicitly depends on $\lambda$) is the one that achieves the smallest prediction error on the test data. When independently generated data is not available, either hold-out data or global cross-validation can be used, which requires coordination of the central machine. If the validation data is kept by the central machine, the central machine can evaluate and compare different values of $\lambda$. If K-fold cross-validation is carried out on the local machines, the central machine needs to send $\bar f$ to the local machines so that the local machines can calculate the cross-validation error, and send the error to the central machine. This needs to be done $K$ times for $K$-fold CV. [1,2] developed generalized approximate cross-validation criterion in the least-squares case and established its asymptotic optimality. We expect that similar criterion developed for quantile regression [3] can be adapted in our case. However, even in the non-distributed setting, [3] did not establish the asymptotic properties of the criterion, mainly investigating its performance from a computational viewpoint. Thus it remains an open question whether such a criterion has a desirable asymptotic property. The investigation of such problems requires additional study by itself and thus we do not use this in the current work. We added discussions in the paper **in the simulation section** with the mentioned references (thank you for providing the references). We have also added more simulation results and real data analysis in the **supplementary appendix**.
>
> ### **Some minor issues**
> Thanks for your careful reading, We have corrected the minor problems you noted.
>
> ### **limitation: negative societal impact and choice of $\alpha$**
>
> The following societal impact statement is added to the conclusion: The simple divide-and-conquer method does not consider the privacy issue in a mathematical rigorous way, for example using the differential privacy concept, which is a concern when using such a simple method.
>
> We mentioned at the end of proof of Theorem 2 (as well the statement of Theorem 2) that for $\bar f$ to achieve the same rate as the non-distributed estimator, we need the constraint  $m\lesssim N^{\min\\{\frac{\alpha^2(2r+1)-(8r+2)\alpha+2r+1}{2\alpha^2(2r+1)+2\alpha},\frac{2\alpha r}{\alpha(2r+1)+1}\\}}$, which leads to the limited choice of $\alpha$. On the other hand, from the proof. we have the convergence rate $N^{-\frac{\alpha(1+2r)}{2\alpha(1+2r)+2}}+(N/m)^{-\frac{\alpha^2}{\alpha^2+4\alpha+1}}$ even if the bound for $m$ is not satisfied (when the bound is not satisfied, we just have non-optimal rate). In other words, we can still allow smaller $\alpha$, just that the convergence rate is now worse. We added a remark to the paper at the **end of Section 2**.
>
> [1] Xu, G., Shang, Z. and Cheng, G., 2018, July. Optimal tuning for divide-and-conquer kernel ridge regression with massive data. In International Conference on Machine Learning (pp. 5483-5491). PMLR.
>
> [2] Xu, G., Shang, Z., and Cheng, G. (2019). Distributed generalized cross-validation for divide-and-conquer kernel ridge regression and its asymptotic optimality. Journal of computational and graphical statistics, 28(4), 891-908.
>
> [3] Yuan, M., 2006. GACV for quantile smoothing splines. Computational statistics & data analysis, 50(3), pp.813-829.
>
> We have also added **more simulation results** (more quantile levels and high-dimensional predictors) and also analyzed the air on-time performance data for illustration. We hope that these responses clarify your questions and concerns. We sincerely welcome you to check our revisions. Please do let us know if you have any follow-up questions or comments!

---

### Official Review · Reviewer_LF2z · 2022-07-11

**Rating:** 5
**Confidence:** 4
**Soundness:** 3 good
**Presentation:** 3 good
**Contribution:** 3 good

**Summary:**

The authors study distributed learning of nonparametric conditional quantiles with Tikhonov regularization in a reproducing kernel Hilbert space (RKHS). The authors establish the risk bounds using a second-order empirical process for quantile risk under the simple divide-and-conquer approach that partitions the data set into multiple parts and then takes an arithmetic average of the individual outputs.

**Questions:**

1. My main concern is in the estimation efficiency when \tau is large/small. When \tau is large/small, the observations located on the tail of the distribution may have a large influence on the estimation results. However, when dividing the sample into multiple groups, such observations may be very few or even totally missing in the subgroup. In other words, the distribution of the subgroup is no longer the same/similar to the full, and the local estimation may be unreliable. How to deal with such cases in the divide-and-conquer approach for quantile regression?

2. Is there any discussion on the number of nodes m in the algorithm?

3. The experiments only consider the one-dimensional case. How about when the dimension of x increases?


**Limitations:**

The work shows the results for \tau=.5 and \tau=.8 for the one-dimensional case. It is better to discuss more cases for more choices of \tau and dimension.

**Strengths And Weaknesses:**

The manuscript includes enough information needed to support the claims it makes. The proposed method is novel in the sense that the authors investigate the theoretical underpinning of the divide-and-conquer strategy for distributed learning of nonparametric quantile regression models.  The method focuses on only an individual value of \tau without modeling different quantiles jointly.

---

> ### Author Response · Authors · 2022-07-31
> **Our point-by-point responses to your review**
>
> Thanks for your great comments and suggestions. We really appreciate the time and efforts you spent in the review. Our point-by-point response are as follows. Note that we have updated the submitted paper according to the comments.
>
>
> ### **efficiency when $\tau$ is small/large**
>
> Thanks for your insightful question. You are right that in practice there is always the problem of estimating extreme conditional quantiles, which is difficult for both the distributed setting and the non-distributed setting. Theoretically, for any fixed $\tau$ in the OPEN interval (0,1), we do not have theoretical problem since we always require $m$ to be much smaller than $N$ and thus typically the local sample size $n=N/M$ is still quite large in the big data setting. Theoretically, the statistical convergence rate inversely depends on the density (this is well-known in the parametric setting, although in the nonparametric setting we are not aware of explicit mentioning of this in the literature). At the tail of the distribution, the density is typically small and thus the statistical error is large, which **can also be seen from our simulations for $\tau=0.1$ and 0.9**. Extreme conditional quantile regression is a separate discipline of its own and has been studied for example in [1]-[3]. However, even in the non-distributed setting we are not aware of any theoretical study of this problem in the framework of RKHS and it remains unclear whether some techniques can be adapted to this setting. We **added discussions on this issue in the conclusion.**
>
> ### **number of nodes**
>
> Thanks for your comments. We mentioned at the end of proof of Theorem 2 (as well the statement of Theorem 2) that for $\bar f$ to achieve the same rate as the non-distributed estimator, we need the constraint  $m\lesssim N^{\min\\{\frac{\alpha^2(2r+1)-(8r+2)\alpha+2r+1}{2\alpha^2(2r+1)+2\alpha},\frac{2\alpha r}{\alpha(2r+1)+1}\\}}$. On the other hand, from the proof. we have the convergence rate $N^{-\frac{\alpha(1+2r)}{2\alpha(1+2r)+2}}+(N/m)^{-\frac{\alpha^2}{\alpha^2+4\alpha+1}}$ even if the bound for $m$ is not satisfied (when the bound is not satisfied, we just have non-optimal rate). In practice, of course it is hard to check whether a specific pair $(m,N)$ satisfies the bound since it s asymptotic in nature. On the other hand, in the setting that data are physically located on $m$ machines and one is unwilling to share data across different machines, then we have no other option and just accept this given $m$. In this situation, we do not have to be concerned with the choice of $m$. We **added a remark to the paper at the end of Section 2**.
>
> ### **more choices of $\tau$ and dimension of $x$**
>
> Thanks for your suggestion. We have added results for $\tau\in\\{0.1,0.2,\ldots,0.9\\}$ with most results in the supplementary appendix. We also added a 3-dimensional example in simulation (very high dimension estimation is not common for nonparametric regression due to curse of dimensionality), and used $\tau\in\\{0.1,0.3,0.5,0.7,0.9\\}$ to make the result less messy and reduce the number of tables, even though this is put in the appendix which have no length constraint. Since there are a lot of new table, please kindly check our revised paper.
>
> [1] Chernozhukov, V. and Du, S. “Extremal quantiles and value-at-risk.” In The New Palgrave Dictionary of Economics (2008).
>
> [2] Wang, H. J. and Li, D. “Estimation of extreme conditional quantiles through power transformation.” Journal of the American Statistical Association, 108(503):1062–1074 (2013).
>
> [3] Wang, H. J., Li, D., and He, X. “Estimation of high conditional quantiles for heavy-tailed distributions.” Journal of the American Statistical Association, 107(500):1453–1464 (2012).
>
> We have also analyzed the **air on-time performance data** for illustration, and added **proof sketches**. We hope that these responses clarify your questions and concerns. We sincerely welcome you to check our revisions. Please do let us know if you have any follow-up questions or comments!

---

### Official Review · Reviewer_TorZ · 2022-07-11

**Rating:** 6
**Confidence:** 4
**Soundness:** 3 good
**Presentation:** 3 good
**Contribution:** 3 good

**Summary:**

This paper establishes the risk bounds for the divide-and-conquer version of the kernel ridge quantile regression estimator when the reproducing kernel has polynomially decaying eigenvalues. To prove the risk bounds, the paper introduces a novel second-order empirical process for quantile risk that can be used to establish a link between the quantile regression estimator and a regularized least-squares retrogression estimator.

**Questions:**

The paper is purely theoretical and introduces a theoretical new device (second-order empirical process) that can be potentially useful for other related problems. However, from a methodological point of view, the paper does not produce any change to the current practice of divide-and-conquer estimation for quantile regression in RHKS.

From a practical point of view, the tuning parameter $\lambda$ is of critical importance and the new theoretical device can potentially be used to derive a data-driven criterion for the selection of $\lambda$, following similar approaches in [1], [2] and [3]. At least some discussions in this direction may strengthen the paper from a practical point of view.

Minor point: Please clarify the meaning of the notation $\mathcal L^r $ in equation (3).

[1] Xu, G., Shang, Z. and Cheng, G., 2018, July. Optimal tuning for divide-and-conquer kernel ridge regression with massive data. In International Conference on Machine Learning (pp. 5483-5491). PMLR.

[2] Xu, G., Shang, Z., and Cheng, G. (2019). Distributed generalized cross-validation for divide-and-conquer kernel ridge regression and its asymptotic optimality. Journal of computational and graphical statistics, 28(4), 891-908.

[3] Yuan, M., 2006. GACV for quantile smoothing splines. Computational statistics & data analysis, 50(3), pp.813-829.

**Strengths And Weaknesses:**

Strengths: the proposed approach is novel by introducing a new theoretical tool that may be useful for other related problems.
Weaknesses: the paper is purely theoretical and lacks practical considerations.

---

> ### Author Response · Authors · 2022-07-31
> **Our point-by-point responses to your review**
>
> Thanks for your favorable comments after reviewing our paper. We sincerely appreciate your time in evaluating the paper, and our point-by-point responses to your comments are given below. We have also uploaded a revised paper.
>
>
> ### **Practical considerations and selection of $\lambda$**
> The main contribution of the current work is the theoretical result, which established the distributed learning rate that under some settings is the same as the rate for the standard centralized estimator. In particular, Theorem 2 shows the choice of $\lambda\sim N^{-\frac{\alpha}{(2r+1)\alpha+1}}$ depends on the total sample size $N$, which indicates that tuning parameter chosen on local data is not optimal. In our simulations, we use independently generated data to select the optimal $\lambda$ that ensure $\bar f$ ($\bar f$ implicitly depends on $\lambda$) is the one that achieves the smallest prediction error on the test data. When independently generated data is not available, either hold-out data or global cross-validation can be used, which requires coordination of the central machine. If the validation data is kept by the central machine, the central machine can evaluate and compare different values of $\lambda$. If K-fold cross-validation is carried out on the local machines, the central machine needs to send $\bar f$ to the local machines so that the local machines can calculate the cross-validation error, and send the error to the central machine. This needs to be done $K$ times for $K$-fold CV. [1,2] developed generalized approximate cross-validation criterion in the least-squares case and established its asymptotic optimality. We expect that similar criterion developed for quantile regression [3] can be adapted in our case. However, even in the non-distributed setting, [3] did not establish the asymptotic properties of the criterion, mainly investigating its performance from a computational viewpoint. Thus it remains an open question whether such a criterion has a desirable asymptotic property. The investigation of such problems requires additional study by itself and thus we do not use this in the current work. We added discussions in the paper **in the simulation section** with the mentioned references (thank you for providing the references). We have also **added more simulation results and real data analysis in the supplementary appendix**.
>
> ### **Meaning of $\mathcal{L}^r$**
>
> Thanks for your comment. Assumption (3) is often referred to as the “source condition” in the literature. $\mathcal{L}^r$ is defined as the operator $\sum_j s_j^r\phi_j\otimes\phi_j$. The condition can be written more explicitly as follows. By Lemma 1.1.1 of Wahba (1990), any $g\in\mathcal{H}$ can be written as $g=\sum_j g_j\phi_j$ with $\sum_j g_j^2/s_j<\infty$, and $\mathcal{L}^r g=\sum_j s_j^rg_j\phi_j$. Thus the assumption is the same as $\sum_j f_{0j}^2/s_j^{1+2r}<\infty$, where $f_{0j}=\langle f_0,\phi_j\rangle$ is the Fourier coefficient of $f_0$ in the expansion. Since $s_j\rightarrow 0$, larger $r$ implies faster decay of $f_{0j}$, and the decay rate of the Fourier coefficient is naturally a characterization of its smoothness in functional analysis. We have **added such remarks after equation (3)**
>
> [1] Xu, G., Shang, Z. and Cheng, G., 2018, July. Optimal tuning for divide-and-conquer kernel ridge regression with massive data. In International Conference on Machine Learning (pp. 5483-5491). PMLR.
>
> [2] Xu, G., Shang, Z., and Cheng, G. (2019). Distributed generalized cross-validation for divide-and-conquer kernel ridge regression and its asymptotic optimality. Journal of computational and graphical statistics, 28(4), 891-908.
>
> [3] Yuan, M., 2006. GACV for quantile smoothing splines. Computational statistics & data analysis, 50(3), pp.813-829.
>
> We have also added **more simulation results** (more quantile levels and high-dimensional predictors) and also analyzed the air on-time performance data for illustration, and added **proof sketches**. We now use the air on-time performance data available We hope that these responses clarify your questions and concerns. We sincerely welcome you to check our revisions. Please do let us know if you have any follow-up questions or comments!

---

> > ### Comment · Reviewer_TorZ · 2022-08-09
> > **response to response**
> >
> > Thank you for your clarification. I am not advocating for developing a distributed version of [3]. My intuition is that since you have developed an approximation of the quantile loss with a weighted least square loss, then a GCV type criterion can be developed using the weighted least square loss in a similar fashion to those in [1,2]. This could significantly strengthen your paper. But I understand this may not be straightforward and I will keep my score as is.

---

> > > ### Author Response · Authors · 2022-08-09
> > > **Thanks for your clarification**
> > >
> > > Thanks again for your comments. Actually the weighted least square is merely used as a proof mechanism. The estimator is not computed via weighted least square because the conditional density is itself harder to estimate than doing quantile regression. Thus it seem directly using weighted least square in extending GCV is not feasible approach. Rather one would probably aim at adapting [3]. Anyway, just a response to your comment and we are not able to provide new information.

---

> > > > ### Comment · Reviewer_TorZ · 2022-08-09
> > > > **Response to response**
> > > >
> > > > I understand that the weighted least square loss may not be directly used to compute GCV based on data, but it can still be used as a theoretical device to derive the specific form and prove the optimality of a distributed version of quantile-GCV. This could be interesting.

---

### Official Review · Reviewer_FH6t · 2022-07-15

**Rating:** 5
**Confidence:** 2
**Soundness:** 2 fair
**Presentation:** 2 fair
**Contribution:** 3 good

**Summary:**


 This work is focus theoretically on the distributed learning of nonparametric quantile regression models in RKHS.

There have been related works about the distributed learning for kernel ridge regression and also the non-distributed learning for quantile regression in RKHS.  The main idea of this paper is to approximate the quantile loss by a weighted least squares loss, then the quantile estimator can be estimated by the weighted least squares estimator.   Based on such approximation,  the authors prove the learning rates for the nonparametric quantile regression learning.

**Questions:**


Q1: The quantile level $\tau\in (0,1)$, however in the illustration section,  there are only results for $\tau = 0.5$ and $\tau =0.8$. Could you please provide more results for the entire range of $\tau$ , say setting $\tau$ 0.1 0.2,...0.8,0.9?

Q2:  What is the performance of this proposed method on real dataset? Quantile regression is a very classical problem and there should be many practical datasets, which have been used in the existing works, to illustrate the distributed quantile regression.


**Limitations:**

yes

**Strengths And Weaknesses:**

The ideas and logic of this paper are very clear and easy to follow. However, the format (especially for all the detailed proofs) seems not formal.The manuscript will benefit from reorganizing the proof details.

---

> ### Author Response · Authors · 2022-07-31
> **Our point-by-point responses to your review**
>
> Thanks a lot for your encouraging words and valuable suggestions. We sincerely appreciate your time in evaluating the paper, and our point-by-point responses to your comments are given below. We have also uploaded a revised paper.
>
> ### **The proof format and presentation**
>
> Thanks for your comments. We have tried to add more details in some parts of the proof, and gives more explanations between theorem and lemmas with a proof sketch added. We hope this can help a little for ease of reading. In particular, the following proof sketches are added:
>
> #### **Sketches of Proof of Theorem 1.** We prove by way of contradiction. Assuming $\\|\widehat f-f_0\\|$ is large, we try to establish
>
> $
> \frac{1}{N}\sum_i\rho_\tau(y_i-\widehat f(x_i))+\lambda\\|\widehat f\\|_\mathcal{H}^2>\frac{1}{N}\sum_i\rho_\tau(y_i-f_0(x_i))+\lambda\\|f_0\\|_\mathcal{H}^2,
> $
>
> which will contradict the fact that $\widehat f$ is the minimizer of the penalized loss. The difference of the quantile loss on the two sides is approximated by its population version with the help of Lemma 1 and bounded below by assumption (A2’), while the difference of the penalties are carefully bounded by a nontrivial application of Young’s inequality for operators, taking into account the source condition.
>
> #### **Sketches of Proof of Theorem 2.** From Lemma 2, $\widehat f_j$ that minimizes the regularized empirical loss can be well-approximated by $\widetilde f_j$ that minimizes the sum of three terms:  $P_n((f(x)-f_0(x))\epsilon)$ (the first-order term in the definition of $g$), the Tikhorov regularization term, and the second-order term from Taylor’s expansion of the *expected* loss. By definition $\widetilde f_j$ is a weighted kernel ridge estimator and the statistical bound for its average over $m$ machines can be more easily established. However, rigorous establishment of the closeness of $\widehat f_j$ and $\widetilde f_j$ requires more detailed analysis involving multiple applications of Lemma 2.
>
> The proof of Lemmas is now moved to the supplementary appendix.
>
> ### **More results for the entire range of $\tau$**
> Thanks for your suggestion. We have added results for $\tau\in\\{0.1,0.2,\ldots,0.9\\}$ with most results in the supplementary appendix. We also added a 3-dimensional example in simulation, and used $\tau\in\\{0.1,0.3,0.5,0.7,0.9\\}$ to make the result less messy and reduce the number of tables, even though this is put in the appendix which have no length constraint. Since there are a lot of new tables, please kindly check our revised paper.
>
> ### **Real dataset**
> Thanks for your suggestion. We now use the air on-time performance data available at https://dataverse.harvard.edu/dataset.xhtml?persistentId=doi:10.7910/DVN/HG7NV7 which consists of flight arrival and departure details for all commercial flights within the USA, from October 1987 to April 2008. The aim is to predict the delay in minutes of the aircraft. After some initial exploration, we use 3 predictors, including ArrivalTime, DeparturetTime and Distance. This is a large dataset, and we randomly select N=10000 observations with another 5000 as validation data for tuning parameter selection. The quantile regression model is fitted to the data with $m\in\\{20,50,100\\}$. We see that the errors decrease with increase of $m$. Because here the true $f_0$ is unknown, the error measure is the average loss for another randomly selected 5000 observations. The whole procedure is repeated 100 times and the results are reported in Table below (standard errors shown in brackets).
>
> | |$\tau=0.1$|$\tau=0.3$|$\tau=0.5$|$\tau=0.7$|$\tau=0.9$|
> |----|----|----|----|----| ---|
> |$m=20$  |  0.890 | 0.361| 0.466| 0.418| 0.863|
> | | ( 0.061 ) | ( 0.022 ) | ( 0.028 ) | ( 0.029 ) | ( 0.059 )|
> |   $m=50$ | 1.229| 0.395| 0.512 |0.479| 0.919|
> | |( 0.073 ) | （0.025 ) | ( 0.032 ) | ( 0.032 ) | ( 0.053 )|
> |   $m=100$ | 1.290| 0.478| 0.477| 0.487| 1.020|
> | |( 0.095 ) | ( 0.036 ) | ( 0.034 ) | ( 0.029 ) | ( 0.061 )|
>
> We hope that these responses clarify your questions and concerns. We sincerely welcome you to check our revisions. Please do let us know if you have any follow-up questions or comments!

---

### Meta-Review · Area_Chair_ZXh3 · 2022-08-27

**Recommendation:** Accept
**Confidence:** Less certain

**Metareview:**

The paper investigates distributed quantile regression for RKHS-based estimators. Unlike in the usually considered case of distributed least squares learning, the quantile setup is more challenging from a technical perspective. Clearly, the strength of this paper is to tackle this challenge.
The weakness of the paper is that it is a technically dense paper that might not be suited for a broader audience, as one of the reviews also suggests. For researchers more familiar with the overall material, however, the paper is nicely written, and for this reason I believe the strength outweighs the weakness. But in any case, a more gentle description of the assumptions plus examples would improve the paper.
Finally, from my personal reading I was surprised that the Bernoulli 2011 paper by Steinwart and Christmann on RKHS-based quantile regression was not mentioned. In particular a comparison of the results seems to be necessary.

In summary, a technically sound paper that should be accepted provided the competition is not too fierce.

**Award:**

No

---

### Decision · Program_Chairs · 2022-09-14

Accept